# Microstructure and Oxidation Behavior of Metal-Modified Mo-Si-B Alloys: A Review

**Laihao Yu, Fuqiang Shen, Tao Fu, Yingyi Zhang ***, **Kunkun Cui, Jie Wang and Xu Zhang**

School of Metallurgical Engineering, Anhui University of Technology, Maanshan 243002, China; aa1120407@126.com (L.Y.); sfq19556630201@126.com (F.S.); ahgydxtaofu@163.com (T.F.); 15613581810@163.com (K.C.); wangjiemaster0101@outlook.com (J.W.); zx13013111171@163.com (X.Z.)
* Correspondence: zhangyingyi@cqu.edu.cn

**Abstract:** With the rapid development of the nuclear industry and the aerospace field, it is urgent to develop structural materials that can work in ultra-high temperature environments to replace nickel-based alloys. Mo-Si-B alloys are considered to have the most potential for new ultra-high temperature structural material and are favored by researchers. However, the medium-low temperature oxidizability of Mo-Si-B alloys limits their further application. Therefore, this study carried out extensive research and pointed out that alloying is an effective way to solve this problem. This work provided a comprehensive review for the microstructure and oxidation resistance of low silicon and high silicon Mo-Si-B alloys. Moreover, the influence of metallic elements on the microstructure, phase compositions, oxidation kinetics and behavior of Mo-Si-B alloys were also studied systematically. Finally, the modification mechanism of metallic elements was summarized in order to obtain Mo-Si-B alloys with superior oxidation performance.

**Keywords:** Mo-Si-B alloys; alloying and modification; microstructure; oxidation resistance; mechanism

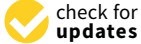



## 1. Introduction

As the most commonly used blade material for aircraft engines, the nickel-based superalloy can operate normally at temperatures below 1150 °C. However, the lower melting point limits its further use. To improve the working efficiency and reduce the fuel consumption of heat engines, the exploration research of new structural materials is promoted. The Mo-Si-B superalloy is considered to be the candidate material with the most potential to replace the nickel-based superalloy because of its outstanding high temperature mechanical properties and excellent oxidation performance [1–8].

In recent decades, Mo-Si-B superalloy has been developed rapidly as the most attractive ultra-high temperature material; however, there are still a large number of problems in the practical industrial application. For example, poor oxidation resistance at medium-low temperatures is one of the non-negligible problems [9–12]. This is because Mo-Si-B superalloys usually contain multiple phases, which exhibit different oxidation behaviors at moderate temperatures. For example, in the low silicon Mo-Si-B alloys (i.e., Si content less than 30 at.%) that contain $Mo_{ss}$-$Mo_5SiB_2$-$Mo_3Si$ phases, $Mo_{ss}$ can improve the fracture toughness of the alloy, but it is easily oxidized and forms volatile molybdenum trioxide in environment temperatures over 350 °C, resulting in significant weight loss of the alloy; $Mo_3Si$ metal compound contains a high content of molybdenum, which leads to poor oxidation resistance [13–17]. For high silicon Mo-Si-B alloys (i.e., Si content more than 30 at.%) containing $MoSi_2$ phase, "pesting oxidation" is a very tricky problem [18–25]. Therefore, the exploration to further improve the antioxidation ability of Mo-Si-B superalloy at moderate temperature has not been interrupted.

Many studies showed that the alloying of metallic elements such as Zr, Al, Fe, and Cr into Mo-Si-B ternary alloys can enhance antioxidation properties by refining the mi-

crostructure or forming the duplex layer, etc. In this work, the oxidation behaviors of low silicon and high silicon Mo-Si-B ternary alloys were reviewed, and studied their oxidation dynamic, microstructure, and oxidation resistance in detail. In addition, this work gave a special and systematic review of the oxidation performance of metal-modified Mo-Si-B alloys, and also analyzed the effects of different metallic elements on the microstructure and oxidation behavior of the Mo-Si-B alloys. Finally, the modification mechanism of the metallic elements was summarized and analyzed to ameliorate the antioxidation performance of the alloy.

## 2. Mo-Si-B Ternary Alloys

The addition of silicon element in the Mo matrix can distinctly enhance the antioxidation capability of the Mo metal at high temperature environment [26,27]. However, the antioxidant performance of the Mo-Si binary alloys is extremely bad at the intermediate temperature (400–800 °C), which often shows "pesting behavior" or accelerated oxidation. The researchers found that a B element can significantly enhance the antioxidation properties of Mo-Si binary alloys [28–31]. Therefore, it is very imperative to discuss the microstructure and oxidation behavior of the Mo-Si-B ternary alloys.

### 2.1. Low Silicon Content of the Mo-Si-B Ternary Alloys

In Mo-Si-B ternary alloys, the silicon and boron elements can form protective borosilicate scale when oxidized on the substrate surface and reduce the oxidation rate of the alloy. Furthermore, the added boron can facilitate the flow of the $SiO_2$ layer on the substrate surface to fill the exterior defects such as cracks and pores, thus enhancing the antioxidation properties of molybdenum metal [32–34]. The microstructure and oxidation performance of Mo-Si-B ternary alloys are closely related to the content of silicon. When the silicon content is low, the research of Mo-Si-B alloys is mostly concentrated at the region of $Mo_{ss}$-$Mo_5SiB_2$-$Mo_3Si$ [35–37]. Wang et al. [38] prepared the Mo-12.5Si-25B (at.%) alloy by arc melting from powder materials of Mo, Si and B, the corresponding microstructure is represented in Figure 1a. EDS and XRD results showed that Mo-12.5Si-25B sample was composed of $Mo_{ss}$ and $Mo_3Si$ implanted into the $Mo_5SiB_2$ matrix. Figure 2a exhibits the oxidation dynamics of this sample at 1200 °C. It is seen that the oxidation dynamics of the alloy consists of two stages: transient oxidation and steady-state oxidation. The mass loss of the alloy is rapid during transient oxidation. This is because the formed borosilicate scale was thin and intermittent at the initial stage of oxidation, which made it difficult to inhibit the rapid evaporation of $MoO_3$. In addition, the gaps provided a channel for the oxygen diffusion into the matrix, which accelerated the oxidation of molybdenum and silicon and formed a Mo-Si-O oxide region at the interface of substrate and scale, as displayed in Figure 1b. As can be observed from Figure 1c, a thicker and continuous oxide film formed on the surface of the alloy when it was oxidized at 1200 °C for 2 h. XRD analysis proved that the scale consisted of silica and dispersed molybdenum dioxide. In addition, an inner layer of $Mo_{ss}$ and diffuse silica was found below the oxide scale. After 2 h of exposure, the oxidation of alloy reached a steady-state stage. However, due to the existence of holes in the alloy (Figure 1c), the oxidation dynamics was still characterized by mass loss, but the loss rate was evidently reduced (Figure 2a). After 100 h of oxidation, a dense and thick oxide scale formed on the surface of the substrate, which was composed of three layers. The outermost layer consisted of silica and a small amount of dispersed molybdenum dioxide, and the inner layer was mainly composed of $Mo_{ss}$ and diffuse $SiO_2$ particles. Combined with EDS and XRD analysis results, there is also a $MoO_2$ layer between the outermost layer and inner layer. Moreover, it can be seen from the local enlarged diagram that the $Mo_3Si$ phase in the interface between the matrix and the inner layer was preferentially oxidized and generated molybdenum and $SiO_2$, as depicted in Figure 1d. Compared with the Mo-12.5Si-25B alloy, the Mo-14Si-28B (at.%) alloy fabricated by arc melting method showed superior antioxidation properties during cyclic oxidation at 1200 °C for 100 h [39,40]. This is because the

Mo-14Si-28B alloy takes only about 2 h to achieve the steady-state oxidation and forms a continuous protective oxide film, which is much earlier than Mo-12.5Si-25B alloy.

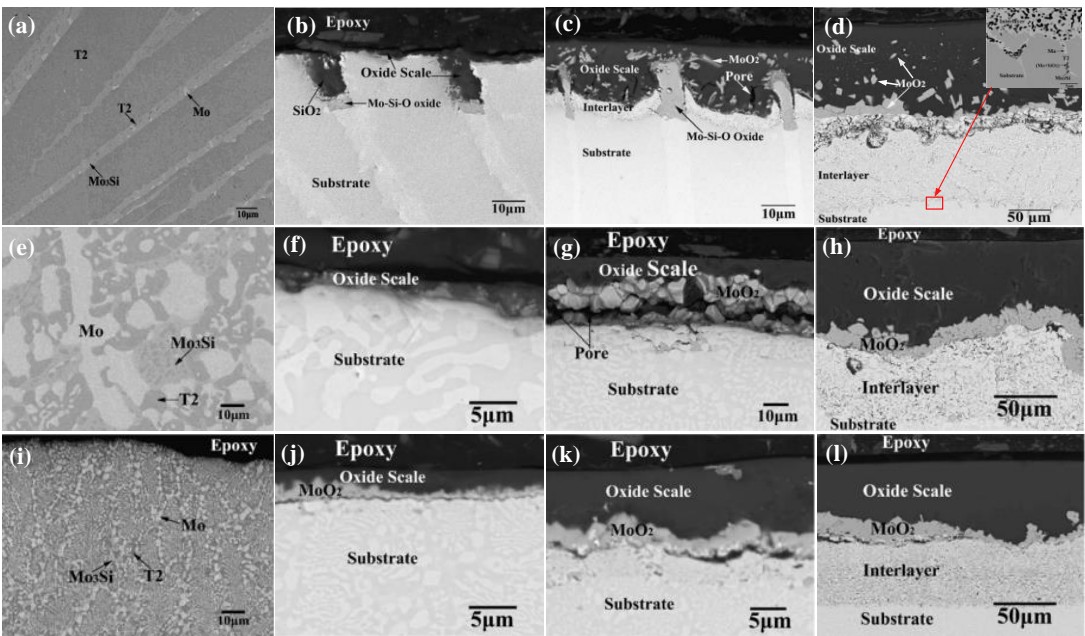

**Figure 1.** BSE micrographs of arc-melted Mo–12.5Si–25B (**a**), annealed (**e**) and laser-remelted Mo–10Si–14B (**i**) samples; Cross-sectional SEM images of the three samples oxidized at 1200 °C for different time: 10 min (**b,f,j**), 2 h (**c,g,k**) and 100 h (**d,h,l**). (**a**–**l**) reproduced with permission [38] and [41], respectively. Copyright 2007 Elsevier.

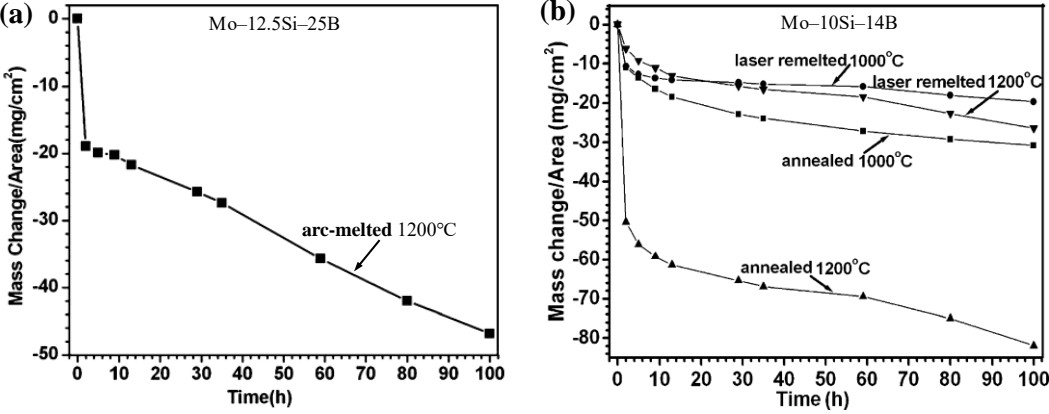

**Figure 2.** Mass change curves: (**a**) arc-melted alloy cyclic oxidation at 1200 °C, (**b**) laser-remelted and annealed alloys oxidized at 1000 °C and 1200 °C. (**a,b**) reproduced with permission [38,41], respectively. Copyright 2007 Elsevier.

In addition, Wang et al. [41] also prepared the Mo-10Si-14B (at.%) alloy through arc melting and studied its oxidation behavior after annealing and laser remelting, respectively. The microstructure of the alloy surface after annealing and laser remelting is shown in Figure 1e,i. The results of EDS and XRD analysis revealed that both alloy surfaces were composed of $Mo_{ss}$, T2 and Mo3Si three phases, but the alloy after laser remelting had a more refined and evenly distributed three phase structure. The cyclic oxidation kinetics of the two alloys at 1000 °C and 1200 °C is presented in Figure 2b. It was found that the mass loss per unit time of the laser remelted was smaller than that of the annealed alloy, and this difference was more obvious at 1200 °C. Therefore, it can be inferred that laser remelting can effectively improve the antioxidation ability of the Mo-10Si-14B sample. To further study their antioxidation mechanism, the cross-sectional SEM images of two treated alloys

after oxidation of 100 h at 1200 °C was analyzed. It can be seen that the annealed alloy formed a thin discontinuity scale consisting of silicon dioxide and dispersed molybdenum dioxide when oxidized for 10 min. After oxidation for 2 h, the scale thickened with many holes. When the oxidation time is 100 h, a relatively continuous scale formed. Moreover, a $MoO_2$ middle layer and an interlayer composed of $Mo_{ss}$ and $SiO_2$ also appeared below the scale, as being presented in Figure 1f–h. By comparison, the laser remelted alloy could form a continuous and uniform $SiO_2$ scale after oxidation for 10 min, and the scale became obviously thicker after oxidation for 2 h. While oxidized for 100 h, the $SiO_2$ scale became denser, as presented in Figure 1j–l. Therefore, it can be inferred that the better antioxidation performance of laser-remelted alloy is attributed to refinement of grain size, which makes the alloy form protective continuous borosilicate scale earlier during oxidation. The studies of Rioult [42] and Choi et al. [43] also confirmed that grain refinement will enhance the antioxidant function of the alloy.

It is reported that boron plays a positive role in improving oxidation protection of the low silicon Mo-Si-B ternary alloys. Because boron can decrease the viscosity of silica film, thus prompting it to flow and cover the surface of the alloy substrate. Li et al. [44] fabricated the 0.9 wt.% $La_2O_3$-doped Mo-12Si-xB (x = 5, 8.5, 17 at.%) samples through mechanical alloying and hot pressing sintering. Figure 3a–c display the microstructures of the three alloys after hot pressure. It can be seen that all the alloy surfaces have fine structure and uniform composition. EDS analysis results of Figure 3a–c show that the bright area is $\alpha$-Mo phase, the gray area is $Mo_3Si$ phase, and the dark gray area is T2 phase, which is consistent with the result of XRD analysis (Figure 4a). Furthermore, the area of bright or gray decreases, while the dark gray area increases significantly, indicating that the volume fractions of $\alpha$-Mo phase and $Mo_3Si$ phase gradually reduce with the raising boron content, while the volume fraction of T2 phase gradually increases. However, the boron content does hardly affect the granularity of each phase, as presented in Figure 4b,c. Figure 3d–f is the TEM bright field micrographs of three alloys. It can be observed that the presence of small $La_2O_3$ particles is inside the grain or on the grain boundary. However, the diffraction peak of the $La_2O_3$ phase is not observed in the XRD pattern (Figure 4a), because its content is too small to be detected. In addition, $La_2O_3$ particles act as "pinning". On the one hand, it can inhibit grain growth, to achieve grain refining effect; on the other hand, it can serve as a barrier phase to inhibit the diffusion of molybdenum ions and the volatilization of $MoO_3$. For the above conclusions, more systematic studies were reported by Zhang [45], Majumda [46,47], and Burk et al. [48,49]. To further study the effect of boron content on the antioxidation behavior of Mo-Si-B alloys, Li et al. [44] also carried out isothermal oxidation experiment of the three samples at 1000 °C and obtained the oxidation dynamics through taking notes the change of mass loss with time during oxidation process, as shown in Figure 4d. It can be found that with the increase of the boron content, the mass loss per unit time of the alloy decreased gradually in the oxidation process, and this phenomenon was the most obvious during the transient oxidation period. Moreover, the Mo-12Si-5B sample had the most weight loss after oxidation for 30 h, showing relatively poor oxidation resistance, which indicated that the increase of boron content played a beneficial role in enhancing the oxidation properties of Mo-Si-B alloys. Figure 5a–d depicts the surface SEM images of Mo-12Si-xB samples when oxidized at 1200 °C for 30 h. On the whole, borosilicate scales were formed on the surface of all Mo-12Si-xB samples, and (Mo, La)-oxide particles with relatively dispersed distribution were observed in this scale. However, after oxidizing Mo-12Si-5B alloy, the borosilicate scale was rough, and numerous cavities, cracks and coarse silica-rich regions appeared on the surface. This is attributed to the comparatively low boron concentration in the alloy that makes it difficult to form a continuous stable borosilicate scale, as shown in Figure 5a,b. As the boron content increased, the borosilicate scale gradually became smooth, continuous and dense, and the cavities, cracks, and coarse silica-rich regions on the alloy surface gradually disappeared, as depicted in Figure 5c,d. However, when the content of boron reaches a certain degree, the continuous increase of boron has little effect on the oxidation rate of the alloy (Figure 4d). Meantime, too much

boron will also lead to the shedding of silica scale due to too low adhesion, which results in poor oxidation resistance. Therefore, increasing the boron content appropriately is of great significance to enhance the antioxidation action of Mo-Si-B alloys.

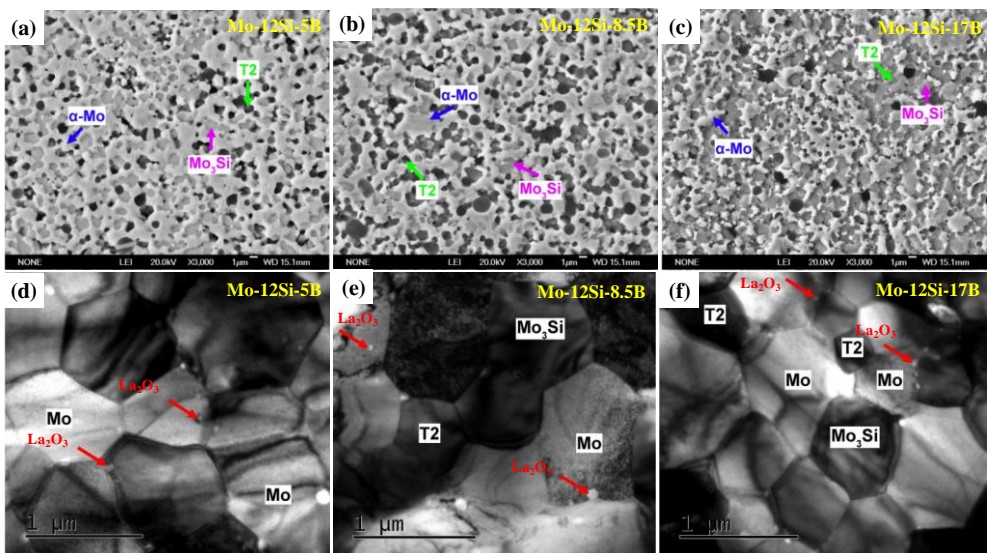

**Figure 3.** Typical SEM (**a**–**c**) and TEM bright field images (**d**–**f**) present the micromorphology of three different alloys. Reproduced with permission [44]. Copyright 2019 Elsevier.

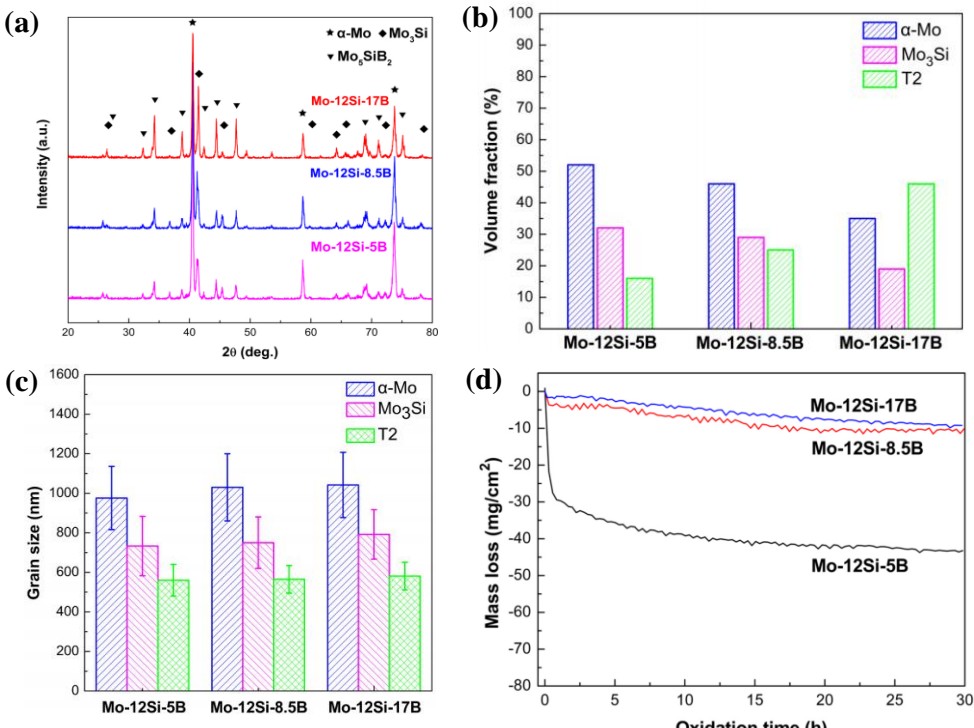

**Figure 4.** (**a**) XRD patterns of different alloys; (**b**) The volume fraction of different phases in three alloys; (**c**) Average grain size of each phase; (**d**) Oxidation behaviors of various alloys at 1000 °C. Reproduced with permission [44]. Copyright 2019 Elsevier.

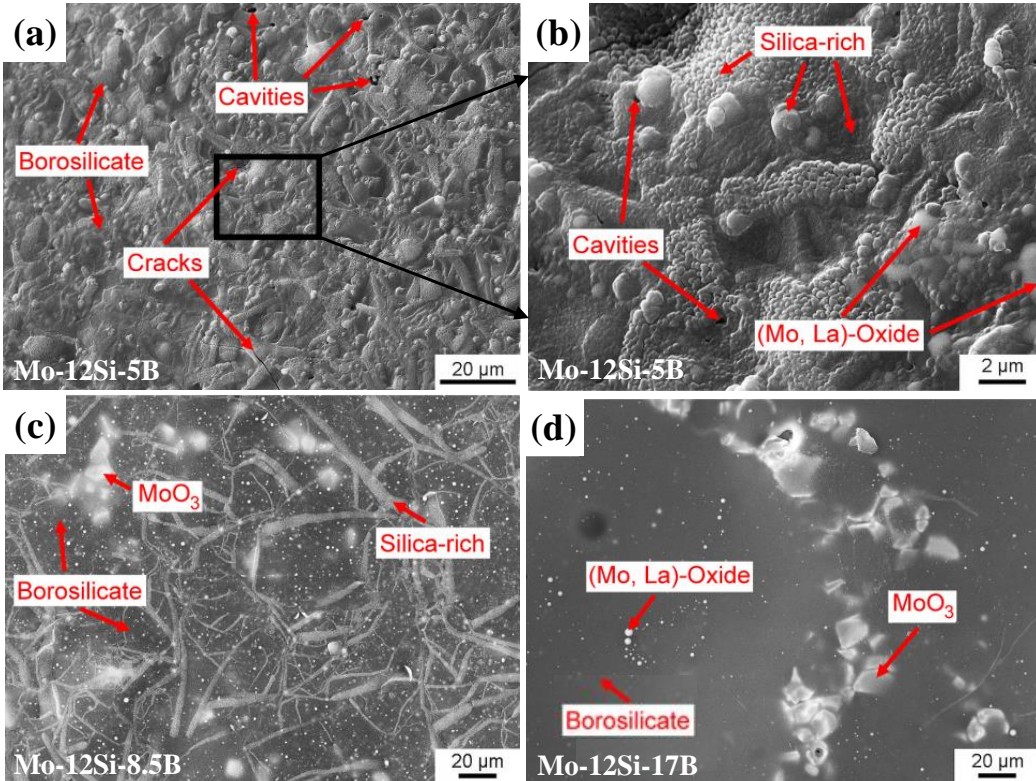

**Figure 5.** Surface SEM images of various alloys during oxidation 30 h at 1000 °C. (**a**) Mo-12Si-5B alloy with magnified image (**b**), (**c**) Mo-12Si-8.5B alloy, (**d**) Mo-12Si-17B alloy. Reproduced with permission [44]. Copyright 2019 Elsevier.

## 2.2. High Silicon Content of the Mo-Si-B Ternary Alloys

In contrast with the low silicon Mo-Si-B alloys, the high silicon Mo-Si-B alloys tend to possess superior antioxidation capacity owing to the high silicon concentration, which is easier to form borosilicate scale covering the substrate surface. For going a step further study the antioxidation performance of high silicon Mo-Si-B alloys, Wen et al. [50] prepared two kinds of Mo-62Si-5B (at.%) alloys with different particle sizes through spark plasma sintering (SPS), using two alloy powders obtained by mechanical disruption (MD) and ball milling (BM) as raw materials. They were also recorded as SPS-MD alloy and SPS-BM alloy, respectively. The typical BSE micrographs of the sintered Mo-Si-B alloys are displayed in Figure 6a,d. XRD analysis results show that the two alloys are composed of MoB phase and $MoSi_2$ phase. However, there are two additional $Mo_5Si_3$ phase and $SiO_2$ phase in SPS-BM alloy, which are unevenly dispersed on $MoSi_2$ matrix, resulting in a fine and dense microstructure. While a large number of holes and cracks appeared on the surface of SPS-MD alloy (Figure 6a), which may lead to relatively poor oxidation resistance. In addition, Wen et al. also investigated the antioxidation function of the two alloys at 1250 °C and 1350 °C respectively, which further confirmed this inference. After oxidation 200 h at 1250 °C or 1350 °C, the oxidation rate of SPS-BM alloy was slower than SPS-MD alloy. In addition, the difference of oxidation rate between the two alloys was more obvious at 1250 °C, which revealed that the SPS-BM alloy had better antioxidant effect, as shown in Figure 6b,c,e,f and Figure 7a,b, which give the cross-sectional images of both alloys when oxidized at 1250 °C and 1350 °C for 30 h, respectively. They point out that with the increase of oxidation temperature, the dimensions of the oxide film becomes thicker, and the change of the oxide layer thickness of SPS-BM is more pronounced. A similar conclusion is also confirmed in Figure 7c,d. Moreover, Figure 7c,d also shows that the oxide layer thickness of SPS-MD alloy is thicker than that of SPS-BM alloy at the same temperature, and the difference of the oxide layer thickness is more obvious at 1250 °C, which is because the

oxidation resistance of SPS-MD alloy is relatively poor. This is consistent with the results of the oxidative dynamics (Figure 7a,b). Furthermore, a lot of cracks and holes were found in the cross-sectional morphology of SPS-MD alloy during oxidation for 30 h, which provided shortcuts for the internal diffusion of oxygen, as shown in Figure 6b,c. No cracks and holes were observed in the cross section of SPS-BM alloy after oxidation, but there were silica particles on the interface between borosilicate scale and substrate. These particles connected borosilicate scale and substrate and played the role of "mechanical locking". Moreover, it promoted the rapid formation of consecutive and protective $B_2O_3$-$SiO_2$ scale on the alloy surface, thus enhancing the antioxidation properties of SPS-BM alloy, as shown in Figure 6e,f. This result was also observed in the study of Pan et al. [51].

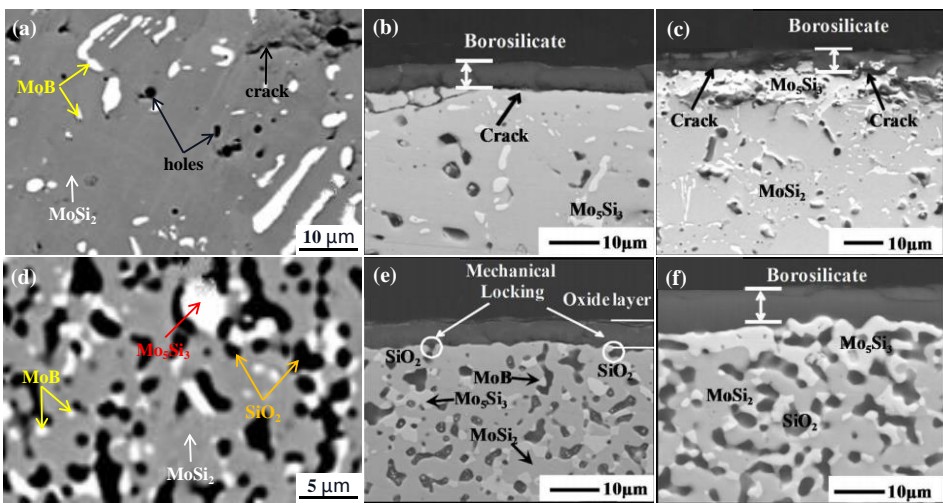

**Figure 6.** BSE micrographs: SPS-MD (**a**), SPS-BM (**d**); Cross-sectional micrographs of SPS-MD and SPS-BM oxidized at different temperatures for 30 h: 1250 °C (**b**,**e**); 1350 °C (**c**,**f**). Reproduced with permission [50]. Copyright 2017 Elsevier.

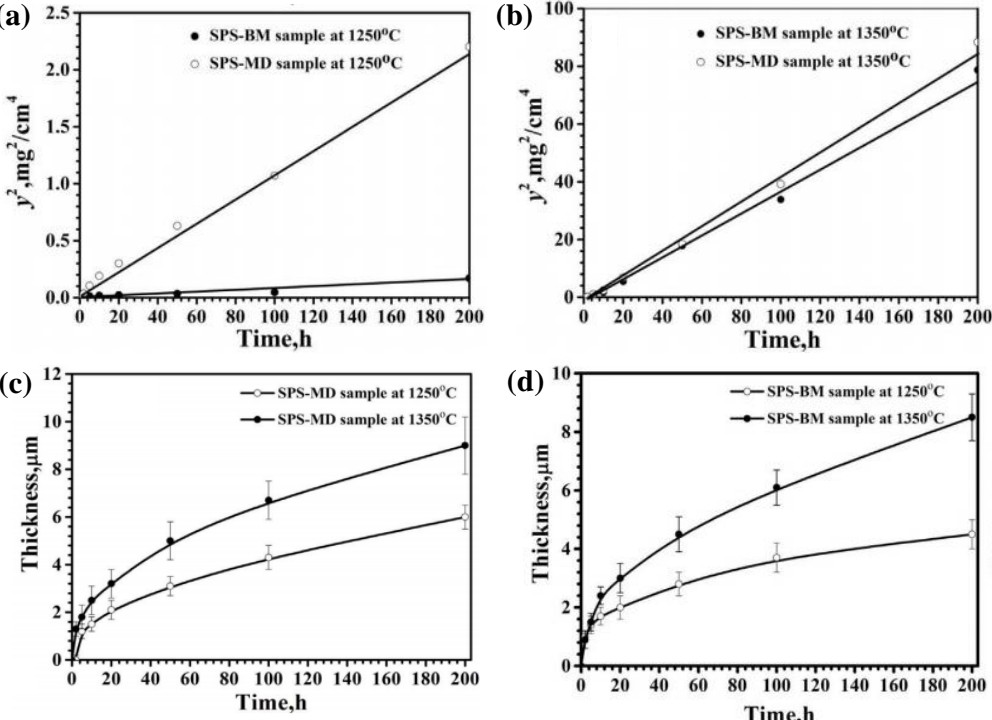

**Figure 7.** (**a**,**b**) Square of mass change ($y^2$) plots at different temperatures; (**c**,**d**) Oxide layer thickness change curve at different temperatures. Reproduced with permission [50]. Copyright 2017 Elsevier.

## 3. Metallic Elements Modified Mo-Si-B Alloys

Although there are significant differences in the oxidation behavior between low silicon and high silicon Mo-Si-B ternary alloys, the antioxidation mechanism of both types of alloys is similar, i.e., the continuous dense $B_2O_3$-$SiO_2$ film is formed on the substrate surface to passivate alloy, so as to strengthen the antioxidation properties of the alloy. However, for Mo-Si-B ternary alloys containing the $Mo_{ss}$ phase, although the $Mo_{ss}$ phase improves the fracture toughness of the alloy, it is likely to be oxidized to volatile molybdenum trioxide even at 350 °C. Furthermore, with regard to Mo-Si-B ternary alloys containing $MoSi_2$ phase, the phenomenon of "pesting behavior" or accelerated oxidation will occur when oxidized at 800 °C [51,52]. Therefore, the improvement of antioxidation function of Mo-Si-B ternary alloys is going to be extensively studied in the following.

### 3.1. Single Metallic Elements Modified Mo-Si-B Alloys

Adding metallic elements such as Zr, Al, and Cr to the Mo-Si-B ternary alloys will refine their microstructure to shorten the period of transient oxidation, and improve the stability of borosilicate scale in the steady oxidation stage. Therefore, it is important to systematically study the Mo-Si-B alloys modified by metal elements.

### 3.1.1. Zr Element Modified Mo-Si-B Alloys

Zr is a kind of active metal element which can refine grain with great application prospect. It cannot only improve the mechanical properties of the alloy, such as strength and ductility, but shorten the formation period of stable continuous $B_2O_3$-$SiO_2$ layer, thus improving the antioxidation properties of alloy. Therefore, it is favored by the majority of researchers. Wang et al. [53] synthesized Mo-12Si-8.5B (at.%, referred to as MSB) and 1.0 wt.% $ZrB_2$-doped MSB (referred to as MSBZ) by mechanical alloying. Figure 8a,e shows the microstructures of MSB and MSBZ alloys. On the whole, the surface composition of the two alloys is uniform and the structure is fine, and they are composed of three regions with different brightness. EDS analysis derived that the bright, gray and dark gray regions in Figure 8a,e were α-Mo, $Mo_3Si$ and $Mo_5SiB_2$, respectively. Moreover, $Zr_{ss}$ particles dispersed in the grain interior or boundary of MSBZ alloy. In addition, compared with MSB alloy, MSBZ alloy microstructure has a finer grain size, as shown in Figure 8i,j. The oxidation kinetics of MSB and MSBZ alloys at 900 °C is displayed in Figure 9a. It can be seen that the net mass variations of MSBZ and MSB after oxidation for 30 h were about +10 mg/cm$^2$ and −180 mg/cm$^2$ respectively, indicating that adding $ZrB_2$ could effectively prevent the weight loss of MSB alloy. It is due to the formation of flat and dense $B_2O_3$-$SiO_2$ film on the surface of MSBZ alloy during the oxidation. Furthermore, lots of white particles were scattered in this scale, as shown in Figure 8f,g. According to the analysis results of EDS (Figure 8k) and XRD (Figure 9b), these white particles were composed of $ZrSiO_4$ or $ZrO_2$. On the one hand, these particles can improve the stability of protective oxide film on the alloy surface. On the other hand, they can be used as diffusion barriers to prevent the inward spread of oxygen or the outward volatilization of $MoO_3$, thus improving the oxidation resistance of the alloy. Similar conclusions were reported by Pan et al. [51]. In contrast, the $B_2O_3$-$SiO_2$ film formed on the MSB's surface during oxidation was uneven, which might be due to the formation of Molybdenum trioxide bubbles at the borosilicate scale. In addition, the MSB's surface also appeared pores, which might be due to the ruptured volatilization of the $MoO_3$ bubbles, as shown in Figure 8b,c. The pores provided a fast path for oxygen inward diffusion to accelerate the oxidation of the alloy. Therefore, the alloy with $ZrB_2$ has a better oxidation resistance compared to the MSB alloy. Furthermore, Wang et al. [54] also studied the antioxidation of MSB and MSBZ alloys at 1300 °C, which further confirmed this conclusion. Figure 9c presents the oxidation kinetics of the MSB and MSBZ alloys at 1300 °C. Generally speaking, the weight variation trend of the two alloys was similar, namely transient oxidation stage and steady oxidation stage. For MSB alloy, the weight loss was serious in the transient stage, and it took a long time (2.7 h) to achieve steady-state. However, the weight variation of the MSBZ alloy in the transient

period was little, and it merely needed three hundred seconds to achieve steady-state. It indicated that MSBZ alloy had a strong oxidation resistance, which was consistent with the results at 900 °C. From the SEM images (Figure 8d,h,l), the smooth and continuous borosilicate scale generated on the surface of MSBZ sample when oxidized for 30 h, and many white particles were embedded in the scale. XRD analysis results (Figure 9d) show that the bright granules were $ZrSiO_4$, $t-ZrO_2$ and $m-ZrO_2$, which were different from the composition of white particles observed at 900 °C (Figure 9b). By comparison, after 30 h of oxidation, many holes and cracks appeared on the surface of MSB alloy, which led to its poor oxidation resistance. This result further demonstrated that the addition of $ZrB_2$ had a positive role in improving the antioxidation of MSB alloy.

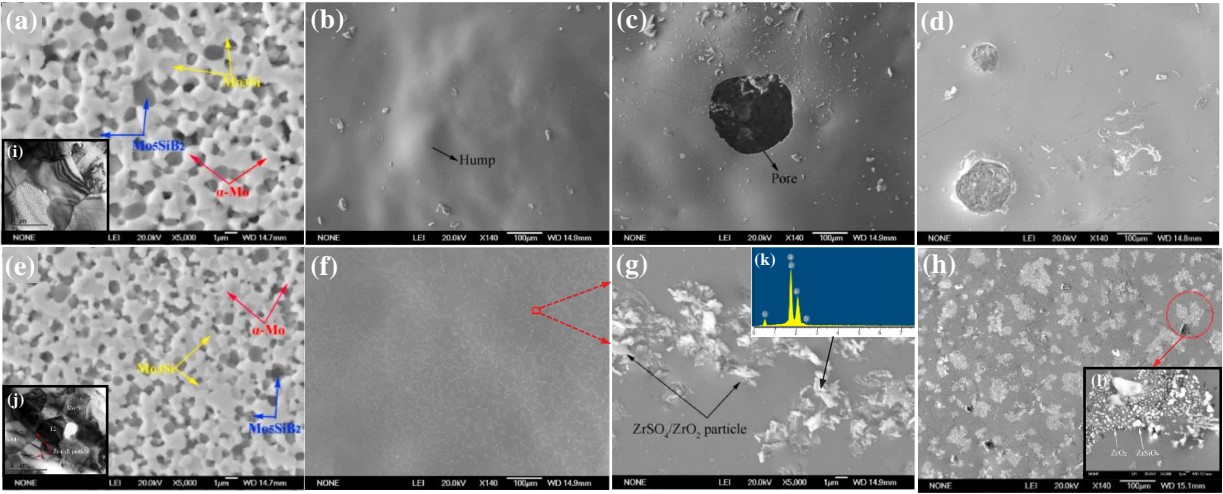

**Figure 8.** SEM micrographs: (**a**) MSB, (**e**) MSBZ with (**i,j**) TEM micrographs; Surfaces of SEM micrographs of the MSB alloy and MSBZ alloy oxidized for 30 h at: (**b,c,f,g**) 900 °C and (**d,h**) 1300 °C; (**k**) EDS pattern of the bright granules in (**g**); (**l**) local magnification image. (**a–c,e–g,k,d,h–j,l**) reproduced with permission [53,54], respectively. Copyright 2017 Elsevier.

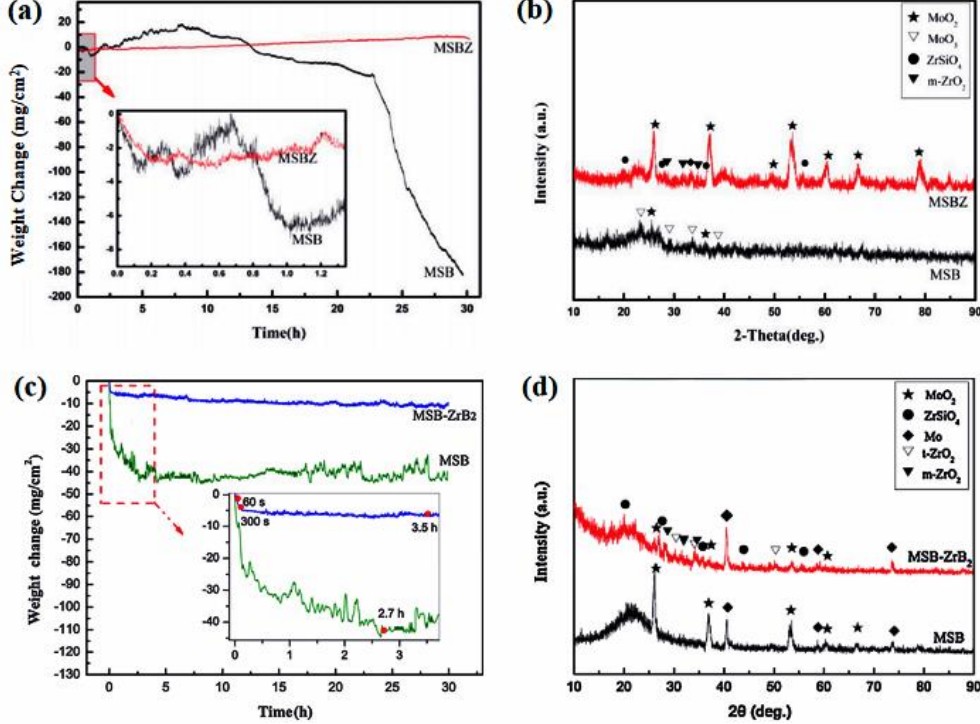

**Figure 9.** The different alloys after oxidation 30 h at (**a,b**) 900 °C and (**c,d**) 1300 °C: (**a,c**) mass change curve, (**b,d**) XRD analysis results. (**a–d**) reproduced with permission [53,54], respectively. Copyright 2017 Elsevier.

In addition, Burk et al. [48,55,56] also systematically researched the effect of adding 1 at.% Zr on the antioxidation capacity of Mo-9Si-8B (at.%) alloy at the medium-high temperature. It was found that the addition of Zr can significantly reduce oxidation rate of Mo-9Si-8B alloy when the temperature is below 1200 °C. In particular, the oxidation speed of Mo-9Si-8B-1Zr (at.%) alloy was nearly three orders of magnitude slower than Mo-9Si-8B sample, which revealed that the addition of Zr was instrumental in improving antioxidation ability of Mo-9Si-8B composite, as depicted in Table 1. However, this beneficial effect of Zr gradually disappeared when the temperature exceeded 1200 °C. Figure 10a displayed the variation curve of the unit area mass with time of Mo-9Si-8B-(1Zr) samples during oxidation at 1300 °C. It could be observed that the oxidation trend of Mo-9Si-8B sample was stable and its weight loss was small. However, the mass change curve of 1 at.% Zr-doped Mo-9Si-8B sample showed a rapid decreasing trend after oxidation for 18 min, which suggested that the antioxidation of Mo-9Si-8B sample had deteriorated after adding Zr. The reason was that when the temperature reached 1300 °C, the phase transformation of $ZrO_2$ in the $SiO_2$ scale occurred, which led to the volume change of $ZrO_2$ particles (expansion or shrinkage), thus destroying the integrity of the $SiO_2$ scale and leaving holes on the alloy surface, providing channels for the volatilization of $MoO_3$ and the inward spread of $O_2$, as shown in Figure 10c,d. In contrast, the surface of the Mo-9Si-8B alloy was overspread with a complete $SiO_2$ scale. Even after oxidation for 72 h, this scale still maintained a continuous and dense structure, as shown in Figure 10b. Therefore, the Mo-9Si-8B composite exhibited superior oxidation resistance at 1300 °C, which was very different from the conclusion of Wang et al. [54]. Kumar et al. [57] argued that this can be caused by the divergences in the B/Si ratio and the microstructural length scales, which made Zr show different effects on the antioxidation of Mo-Si-B alloys.

**Table 1.** The oxidation rate constants ($kg^2/m^4s$) of Mo–9Si–8B–(1Zr) at 1000–1200 °C. Reproduced with permission [55]. Copyright 2009 Springer Nature.

| Temperature (°C) | Oxidation Rate Constants ($kg^2/m^4s$) | |
|:---:|:---:|:---:|
| | **Mo-9Si-8B** | **Mo-9Si-8B-1Zr** |
| 1000 | $1.0 \times 10^{-9}$ | $2.78 \times 10^{-12}$ |
| 1100 | $5.85 \times 10^{-11}$ | $9.0 \times 10^{-12}$ |
| 1150 | - | $1.22 \times 10^{-10}$ |
| 1200 | $1.2 \times 10^{-8}$ | $3.0 \times 10^{-8}$ |

### 3.1.2. Ti Element Modified Mo-Si-B Alloys

The effect of Ti on the properties of Mo-Si-B alloys has been widely studied. On the one hand, the addition of Ti can make Mo-Si-B alloys have lower density and superior creep resistance. On the other hand, Ti can also act as a stabilizer for $Mo_5SiB_2$ and $Mo_5Si_3$ phases. Schliephake [58] and Azim et al. [59] further explored the impact of adding Ti on the oxidation resistance of the Mo-9Si-8B (at.%) alloy based on the Burk [48] research. By comparing the isothermal oxidation weight change curves of Mo-9Si-8B-(29Ti) (at.%) composites at 1100 °C to 1300 °C, it was found that the weight loss of both alloys increased with the rise of oxidation temperature. In addition, the mass wastage of Mo-9Si-8B-29Ti composite was more serious than Mo-9Si-8B composite, indicating that adding Ti did not enhance the antioxidation of Mo-9Si-8B composite, as depicted in Figure 11. To further analyzed the causes of this result, the surface and cross-sectional microstructure of the 29 at.% Ti-doped Mo-9Si-8B composite before and after oxidation were deeply studied. Unlike the microstructural composition of the Mo-9Si-8B alloy, the Mo-9Si-8B-29Ti alloy was mainly composed of $Mo_{ss}$, $Mo_5SiB_2$ and $(Ti, Mo)_5Si_3$, as being displayed in Figure 12a. After oxidation 100 h at 1100 °C, the rutile film formed on the surface of Mo-9Si-8B-29Ti composite, and lots of unevenly dispersed silica was embedded in the scale. In addition, the existence of microcracks was also observed from the scale enlarged diagram. With the increase of oxidation temperature, these microcracks grew and expanded, thus forming

many gaps between or within the ruite particles. It may be related to the expansion of large oxides (TiO$_2$) and the release of internal stress, as shown in Figure 12b,c,f. As previously mentioned, the excellent antioxidation of Mo-9Si-8B composite at temperatures above 1100 °C was caused by the formation of continuous and dense oxide scale (Figure 10b). By contrast, the oxide layer structure of 29 at.% Ti-doped Mo-9Si-8B composite was completely different. When the Ti-doped sample was oxidized at 1100 °C for 100 h, a thick and porous oxide layer formed. The outermost layer was rutile layer, which was mostly composed of TiO$_2$ and little embedded borosilicate. However, the rutile scale had no protective effect, thus it could not serve as barriers against the inward spread of oxygen. Directly below the rutile layer was the TiO$_2$-borosicate duplex layer. However, there were many pores in the duplex layer, which provided a fast route for oxygen diffusion into the substrate and accelerated the alloy oxidation, as described in Figure 12d,g,h. Moreover, With the rise of oxidation temperature, the oxide layer of Mo-9Si-8B-29Ti alloy became thicker and more porous. The author believed that it was attributed to the accelerated growth of TiO$_2$ and the mismatched thermal expansion coefficients between TiO$_2$ and SiO$_2$, as presented in Figure 12e. Therefore, the addition of Ti to the Mo-9Si-8B alloy destroyed the integrity of the borosilicate scale, resulting in its poor oxidative resistance. Schliephake et al. [60] also reached the same conclusion.

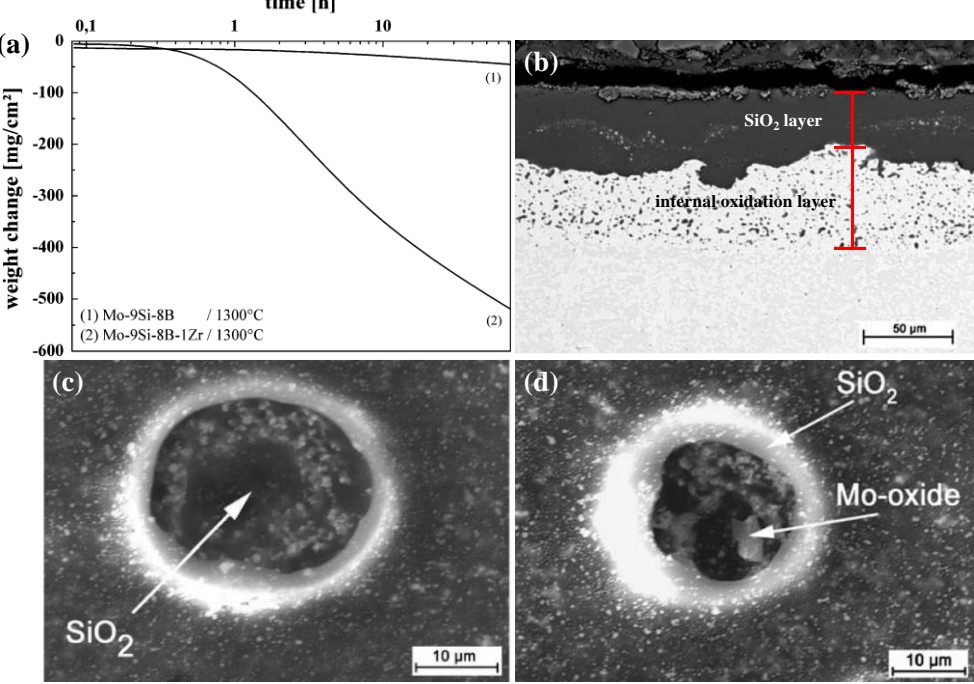

**Figure 10.** (**a**) Weight change curve of both samples at 1300 °C; (**b**) Cross-sectional SE image of Mo–9Si–8B sample after oxidation 72 h at 1300 °C; (**c**,**d**) SE images of Mo–9Si–8B–Zr sample surface after oxidation 15 min at 1300 °C. Reproduced with permission [55]. Copyright 2009 Springer Nature.

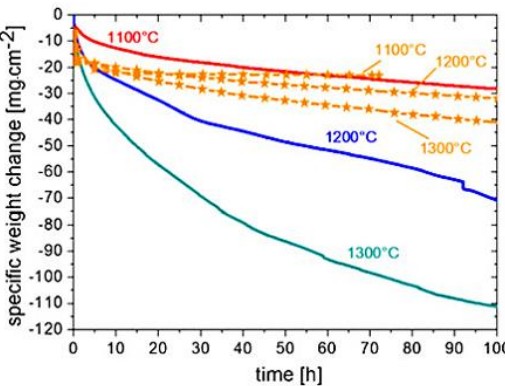

**Figure 11.** Specific weight change vs. time for the Ti-doped (continuous curves) and Ti-free (discontinuous curves) Mo-9Si-8B composites oxidized at various temperatures. Reproduced with permission [58]. Copyright 2013 Springer Nature.

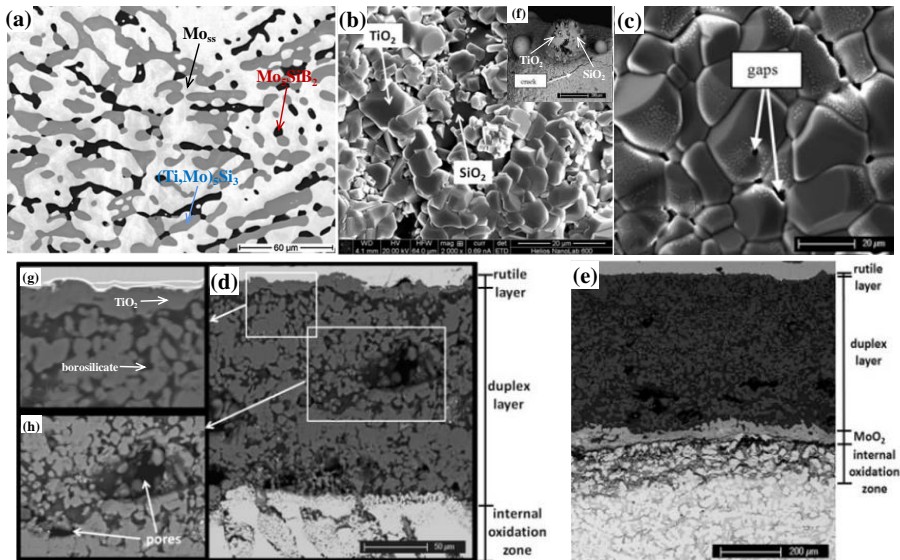

**Figure 12.** SEM (BSE) images of Mo-9Si-8B-29Ti composite: (**a**) microstructure, (**b,c,f**) surface and (**d,e,g,h**) cross section after oxidation 100 h at (**b,d,f–h**) 1100 °C, (**c**) 1200 °C and (**e**) 1300 °C. (**a–h**) reproduced with permission [58,59], respectively. Copyright 2013 Springer Nature.

### 3.1.3. Al Element Modified Mo-Si-B Alloys

Al element modified Mo-Si-B alloys can improve the microstructure and form a continuous protective $Al_2(MoO_4)_3$ scale on the outer surface of borosilicate film, therefore improving their antioxidation capacity. On the other hand, the added Al competes with Si, B and Mo for oxidation and brings about the formation of protective $B_2O_3$-$SiO_2$ film more slowly, which makes the oxidation resistance of the alloy worse [61–63]. This characteristic of Al makes it show different effects on antioxidation of the Mo-Si-B alloys. Paswan et al. [64–66] carried out isothermal and non-isothermal oxidation tests about Mo-14Si-10B (at.%) alloy and 7.3 at.%. Al-doped Mo-11.2Si-8.1B (at.%) alloy fabricated by reaction hot pressing, and recorded the mass variations of both alloys after oxidation at 400–1300 °C for 24 h respectively, as shown in Table 2. It was seen that the net mass change value of the Mo-11.2Si-8.1B-7.3Al alloy was more than the Mo-14Si-10B alloy at each temperature, which revealed that the addition of the Al reduced the oxidative resistance. In the study of the alloy microstructure, it was found that the two alloys presented multi-phase microstructures, and WDS analysis concluded that the white, gray, dark gray and black regions were Mo, $Mo_3Si$, T2 and $SiO_2$ phases, respectively, as being shown in Figure 13a,c. The $SiO_2$ might be produced by the alloy powder during hot pressing. Moreover,

there was an additional $Al_2O_3$ phase in the black region of the Mo-11.2Si-8.1B-7.3Al alloy microstructure. These $Al_2O_3$ particles hindered protective borosilicate scale coverage of the alloy surface, providing a pathway for inward diffusion of oxygen. Moreover, with the increase of oxidation temperature, aluminum borate or mullite might also be precipitated in the borosilicate scale on the surface of Mo-11.2Si-8.1B-7.3Al alloy [67], which further inhibited the passivation of the alloy. Figure 13b,d depicted the surface morphology of Mo-14Si-10B alloy and Mo-11.2Si-8.1B-7.3Al alloy after isothermal oxidation at 700 °C for 24 h, respectively. It served to show the surface oxide layer of Mo-11.2Si-8.1B-7.3Al alloy was rougher and existed a large number of holes, which might be related to thermal chock and mismatch of the thermal expansion coefficient of various oxides in the oxide layer. Therefore, the addition of Al was detrimental to the oxidation resistance of the alloy. However, Yamauchi et al. [68] drew a different conclusion in the study of antioxidation performance of the Mo-Si-B alloy. They observed that adding Al could promote the quick formation of protective Al-Si-O oxide layers on the surface of Mo-Si-B sample between 800 °C and 1300 °C, thus effectively ameliorating its oxidative resistance. The difference in these research results may be caused by the different preparation methods and composition of the alloys.

**Table 2.** Mass loss of both alloys after oxidation at different temperatures for 24 h. Reproduced with permission [64,65]. Copyright 2006 Elsevier and 2007 Elsevier.

| S. No. | T (°C) | Mass Loss (mg/cm$^2$) | |
| --- | --- | --- | --- |
| | | **Mo-14Si-10B** | **Mo-11.2Si-8.1B-7.3Al** |
| 1 | 500 | +0.6 | −0.7 |
| 2 | 600 | +1.2 | −4 |
| 3 | 700 | −92.43 | −369.45 |
| 4 | 800 | −36.97 | −677.43 |
| 5 | 900 | −18.83 | −621.54 |
| 6 | 1150 | −1.11 | −131.44 |
| 7 | 1300 | −112.81 | −192.37 |

### 3.1.4. W Element Modified Mo-Si-B Alloys

In the Mo-Si-B ternary alloys, the presence of $Mo_3Si$ can neither enhance the oxidation resistance nor improve the fracture toughness. Therefore, the removal of the $Mo_3Si$ phase has a positive effect on improving the properties of the alloy. Ray et al. [69] found that adding tungsten to the Mo-Si metal compound would destabilize the $Mo_3Si$ structure. It is worth mentioning that adding tungsten to Mo-Si-B alloys can also remove the brittle $Mo_3Si$ phase and help to form the desired $Mo_{ss}$, $Mo_5SiB_2$ and $Mo_5Si_3$ phases, which will improve the stability, fracture toughness and oxidation resistance of the alloy. Based on this property of tungsten, Karahan et al. [70] further discussed the influences of tungsten addition on microstructure and antioxidation performance of the Mo-Si-B alloys. They prepared Mo-15Si-15B (at.%) and Mo-15W-15Si-15B (at.%) alloys using drop-casting, which were abbreviated as W0 and W15, respectively. The microstructures of the two alloys obtained by casting are shown in Figure 14a,b. It can be observed that there were three phases in the microstructure of W0 alloy, namely metal rich phase in the white region, A15 phase in the gray region and T2 phase in the black region, respectively. In contrast, there were metal rich phase (white region), T2 phase (($Mo$, $W)_5SiB_2$, gray region) and T1 phase (($Mo$, $W)_5Si_3$, black region) in the microstructure of W15 alloy. However, the presence of the A15 phase was not observed, which might be because adding tungsten destroyed stability of A15 and made it form the T1 phase. Figure 15a depicted the mass change curve per unit area of alloys with different W content oxidized at 1200 °C. It can be found that compared with the W0 alloy, the mass loss of the alloy with the addition of W was more serious, and the mass loss of the W15 alloy was the most serious, which indicated that the addition of tungsten was harmful to the antioxidation of the alloy. It was because at 1200 °C, the radioactive $WO_3$ generated by oxidation was unevenly dispersed in the

borosilicate scale and destroyed the composition of the alloy [71–73], resulting in $B_2O_3$-$SiO_2$ film discontinuity, as being depicted in Figure 14c. The discontinuous scale could not serve as barriers to the inward spread of $O_2$, so the antioxidation of W15 alloy became worse. However, as the oxidation temperature was further elevated, $WO_3$ gradually volatilized (Figure 15b). It was noteworthy that when the temperature exceeded 1400 °C, $WO_3$ in the borosilicate scale was almost exhausted by volatilization, and the oxide film became dense (Figure 14d). Therefore, the addition of W might lead to poor antioxidation performance of Mo-Si-B alloys, but it would not cause significant deterioration. Moreover, it also suggested that the pretreatment of Mo-Si-B-W composites at higher temperatures (>1400 °C) can effectively improve the oxidation protection effect.

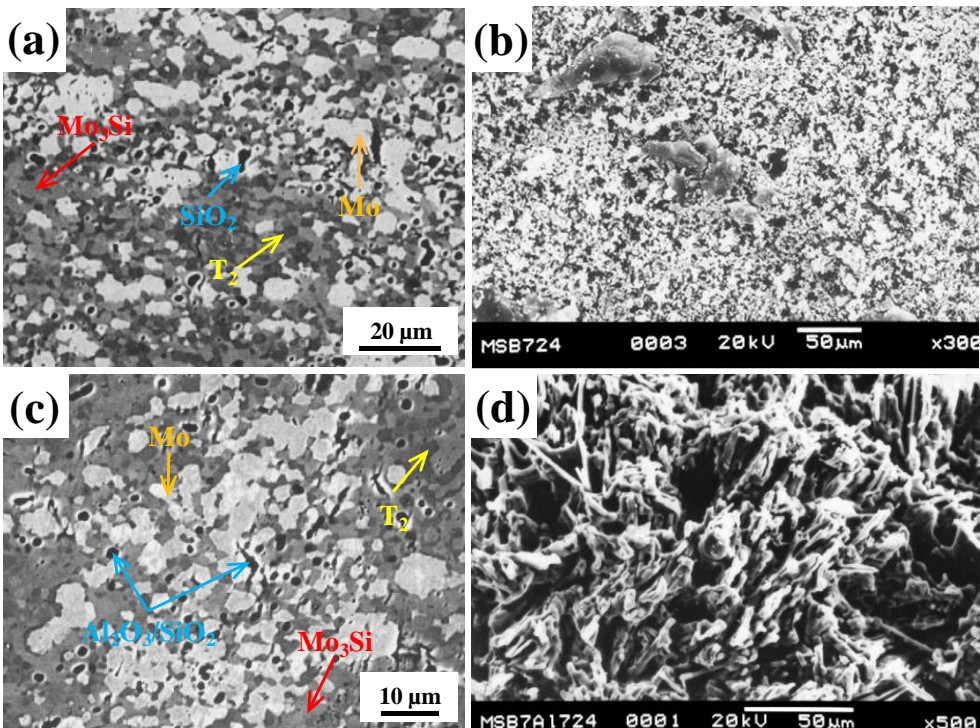

**Figure 13.** BSE images of the Mo-14Si-10B alloy (**a**) and Mo-11.2Si-8.1B-7.3Al alloy (**c**) before the oxidation; Surface SE images of Mo-14Si-10B alloy (**b**) and Mo-11.2Si-8.1B-7.3Al alloy (**d**) after oxidation at 700 °C for 24 h. (**a**–**d**) reproduced with permission [64,65], respectively. Copyright 2006 Elsevier and 2007 Elsevier.

### 3.1.5. Other Single Metallic Elements Modified Mo-Si-B Alloys

Similar to tungsten action, adding niobium to Mo-Si compounds can also destroy the stability of A15 structure [74]. In addition, niobium has a low density and is easy to dissolve in Mo, thus reducing the density of molybdenum-based alloys and making it exhibit excellent high temperature strength and fracture toughness. In recent years, researchers have carried out lots of experiments about improving the properties of Mo-Si-B alloys by niobium. Yang et al. [75] produced Nb-free and 26 at.% Nb-doped Mo-12Si-10B (at.%) alloys through mechanical alloying and hot pressing, abbreviated to 0 Nb and 26 Nb alloys, respectively. They found that the mechanical behaviors such as fracture toughness, compactibility and compression strength of 0 Nb alloy were significantly improved after adding Nb. However, the oxidation resistance of the alloy became worse. Figure 16a gave the oxidation kinetic functions of the 0 Nb and 26 Nb alloys at 1300 °C. It was seen that the 26 Nb alloy lost more mass after oxidation for the same time. Furthermore, it was also observed from the macroscopic image that the 26 Nb alloy was severely destroyed after oxidation for 5 h. In contrast, the 0 Nb alloy still retained the original shape, which further indicated that adding Nb could reduce the antioxidation of the alloy. It may be associated

with non-protective porous oxides formed on the 26 Nb alloy's surface. In addition, XRD and EDX records suggested that these oxides were $Nb_2O_5$ (Figure 16b). Furthermore, the dispersion of $Nb_2O_5$ on the surface of the alloy caused the silica scale to become loose and porous, which provided a fast channel for the diffusion of oxygen into the substrate, as being shown in Figure 17a. Behrani et al. [76] also reported a similar conclusion.

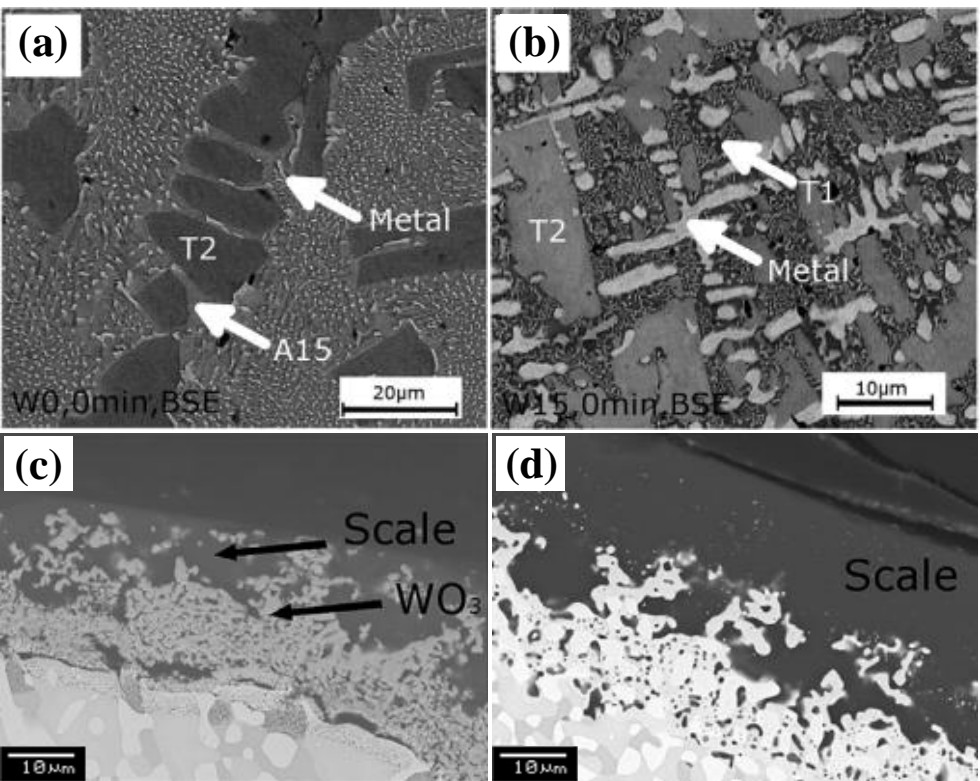

**Figure 14.** BSE images of the W0 (**a**) and W15 (**b**) surface prior to oxidation; Cross-sectional images of W15 after oxidation at 1200 °C (**c**) and 1500 °C (**d**) for 2 h. Reproduced with permission [70]. Copyright 2017 Elsevier.

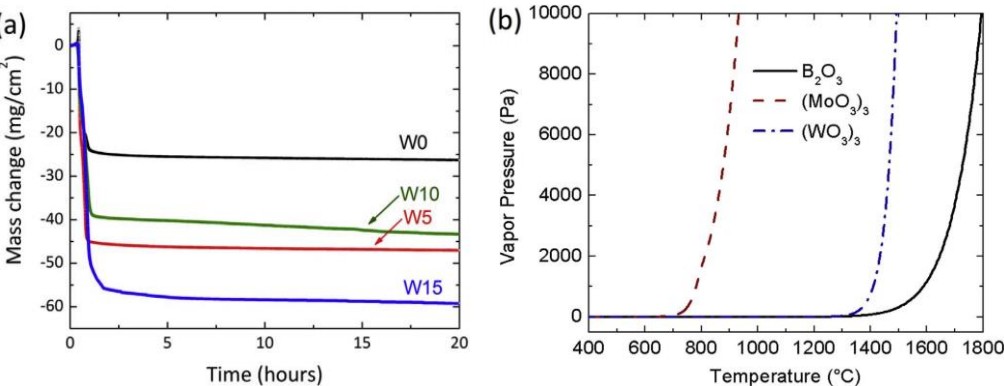

**Figure 15.** (**a**) Weight change curve of all samples at 1200 °C; (**b**) Variation of vapor pressure with temperatures. Reproduced with permission [70]. Copyright 2017 Elsevier.

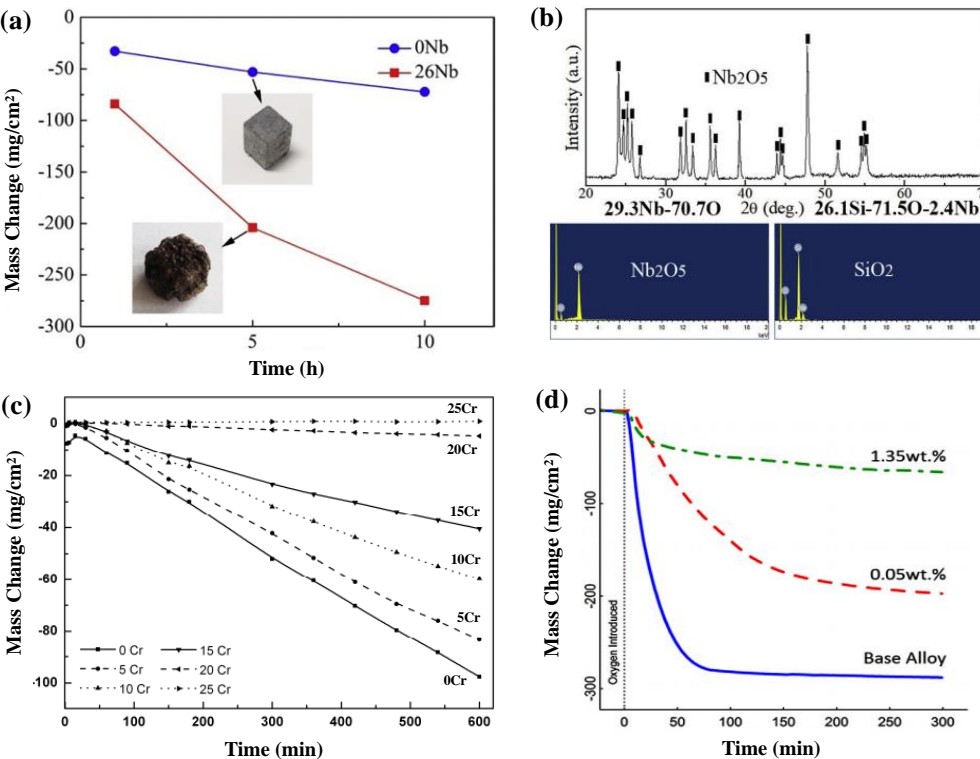

**Figure 16.** Mass change curves: (**a**) 0 Nb and 26 Nb after oxidation at 1300 °C with macroscopic images of samples oxidized for 5 h, (**c**) Mo-9Si-8B samples with different amounts of Cr oxidized at 750 °C, (**d**) Mo–2Si–1B samples with different amounts of Fe oxidized at 1100 °C; (**b**) XRD and EDX results of 26 Nb oxidized at 1300 °C for 5 h. (**a–d**) reproduced with permission [75,77,78], respectively. Copyright 2019 Elsevier, 2011 Springer Nature and 2012 Elsevier.

Compared with Nb, traditional metallic elements (like Cr, Fe) have a positive impact on the oxidative protection of Mo-Si-B ternary alloys, so they are favored by many researchers. The oxidation behavior of Cr modified M-Si-B alloy was relatively systematically explored by Burk et al. [49,77]. Figure 16c shows the mass change curve of Mo-9Si-8B (at.%) alloy containing different amounts of chromium when oxidized at 750 °C. It can be seen that the addition of Cr reduces the mass wastage of Mo-9Si-8B sample, and with the increase of Cr content, its weight loss gradually decreases. When the Cr content reached 25%, the alloy experienced little weight loss, showing excellent oxidative resistance. This was because the relatively high content of chromium made the alloy surface form a stable continuous Cr-oxide scale (consisting mainly of $Cr_2(MoO_4)_3$) that acted as a diffusion barrier of oxygen to passivate the alloy and prevent further oxidation, as being shown in Figure 17b. Moreover, Sossaman et al. [78] researched the antioxidation performance of Mo-2Si-1B (wt.%) composite at 1100 °C. In addition, they found that adding Fe can observably reduce the mass loss of the Mo-2Si-1B composite during transient oxidation process, especially after adding 1.35 wt.% Fe, the mass loss of the alloy even reduced by 75%, as displayed in Figure 16d. It was due to the fact that Fe reduced the stickiness of borosilicate film, thus facilitating the flow of oxide film and making it quickly cover the alloy surface to form a continuous scale (Figure 17d). In contrast, the Mo-2Si-1B alloy failed to form continuous borosilicate film during transient oxidation process (Figure 17c), so $MoO_3$ volatilized rapidly in the early oxidation stage, resulting in severe mass loss. In addition, a large number of $Fe_2(MoO_4)_3$ particles attached to the surface of the Mo-2Si-1B-1.35Fe (wt.%) alloy, which were filled in the pores of borosilicate scale and prevented the diffusion of oxygen. Therefore, the addition of Fe can significantly enhance the antioxidation properties of the alloy, and the same results were reported by Woodard et al. [79].

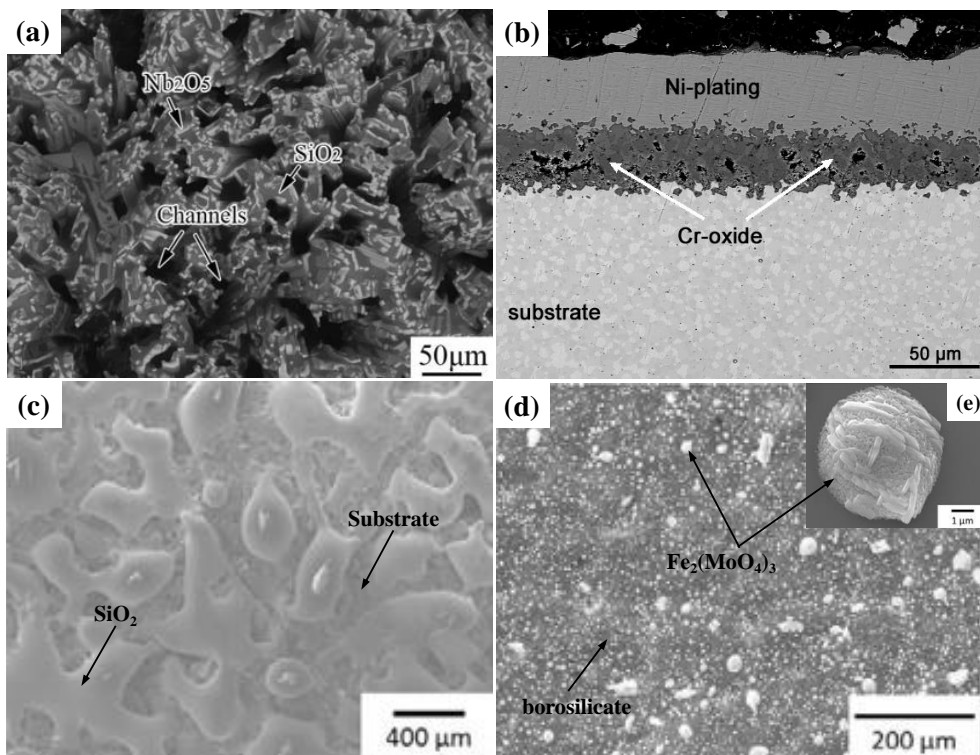

**Figure 17.** Surface SEM-BSE images of 26 Nb oxidized at 1300 °C for 5 h (**a**); Cross section of Mo-9Si-8B-25Cr after oxidation 10 h at 750 °C (**b**); SEM images of both samples after oxidation 1 min at 1100 °C: Mo–2Si–1B (**c**) and 1.35 wt.% Fe-doped Mo–2Si–1B (**d**) with a high magnification image (**e**). (**a**–**d**) reproduced with permission [75,77,78], respectively. Copyright 2019 Elsevier, 2011 Springer Nature and 2012 Elsevier.

### 3.2. Multiple Metallic Elements Co-Modified Mo-Si-B Alloys

In contrast with single metallic elements modified Mo-Si-B alloys, the modification mechanism of multiple metallic elements tends to be more complex. It has been previously described that thick and porous TiO2-borosilicate duplex scale was generated on the surface of Mo-Si-B composites after adding Ti, which led to rapid oxidation of the alloys. To further enhance the antioxidant capacity of Mo-Ti-Si-B composites, Zhao et al. [80] again added the fifth metallic element, such as Al or Cr, and synthesized the 35Mo-35Ti-20Si-10B (at.%), 35Mo-35Ti-15Si-10B-5Al (at.%) and 32.5Mo-32.5Ti-20Si-10B-5Cr (at.%) alloys, which were referred to as BASE, AL and CR, respectively. The microstructures of three samples are shown in Figure 18a–c. It was observed that the microstructures of the three alloys were composed of $Mo_{ss}$ (white region), $Mo_5SiB_2$ (gray area) and $Ti_5Si_3$ (dark gray region). However, the addition of Al or Cr affects the morphology or volume fraction of individual phase. For example, adding Al refines the structure of the $Mo_{ss}$ and $Ti_5Si_3$ phases (Figure 18b), and the addition of Cr reduces the volume fraction of the $Mo_{ss}$ phase (Figure 18c). Therefore, BASE, AL and CR alloys can exhibit different oxidation behaviors at the same temperature. Figure 19a,b gives the change of unit area weight with oxidation time of the three alloys at 700 °C and 1100 °C, respectively. Overall, the weight loss of AL and CR alloys was lower than the BASE alloy at the both oxidation temperatures, which indicated that adding Al or Cr can improve the antioxidation function of the BASE sample. This was related to AL or CR alloy with finer $Mo_{ss}$ and $Ti_5Si_3$ structures and lower $Mo_{ss}$ volume fraction. It was worth noting that CR alloy had better antioxidant than AL and BASE alloys at 1100 °C (Figure 19b). For a deeper understanding of the enhancement mechanism of adding aluminum or chromium to the antioxidation of Mo-Ti-Si-B alloys, the surface and section of the postoxidation alloys were studied. Figure 18d,e are the surface SEM micrographs of AL and CR alloys during oxidation for 30 min at 700 °C, respectively.

It can be found that the surface borosilicate scales of AL and CR alloys had low viscosity and were easy to flow, which made the AL and CR alloys form the $TiO_2$-borosilicate duplex scale covering the substrate surface earlier than the BASE alloy. Therefore, the antioxidation of the AL and CR alloys were better than that of the BASE alloy at 700 °C. Compared to the cross-sections of three samples oxidized at 1100 °C for 24 h (Figure 18f–h), it can be seen that lots of holes and cracks appeared in the duplex scale of BASE and AL alloys, which provided a way for inward diffusion of oxygen. In contrast, continuous protection films are generated on the surface of CR alloy, and no holes appeared in the duplex scale. Therefore, the CR alloy possessed optimum oxidative resistance at 1100 °C. It was worth mentioning that there was no continuous Cr-oxide scale formed on the surface of CR sample during oxidation (Figure 18i). Similarly, there was no continuous Al-oxide scale on the surface of AL sample. This may be caused by the small content of Al or Cr in the samples. Therefore, it can be envisaged that to further enhance the antioxidation of alloys can be realized by increasing the concentration of Al or Cr.

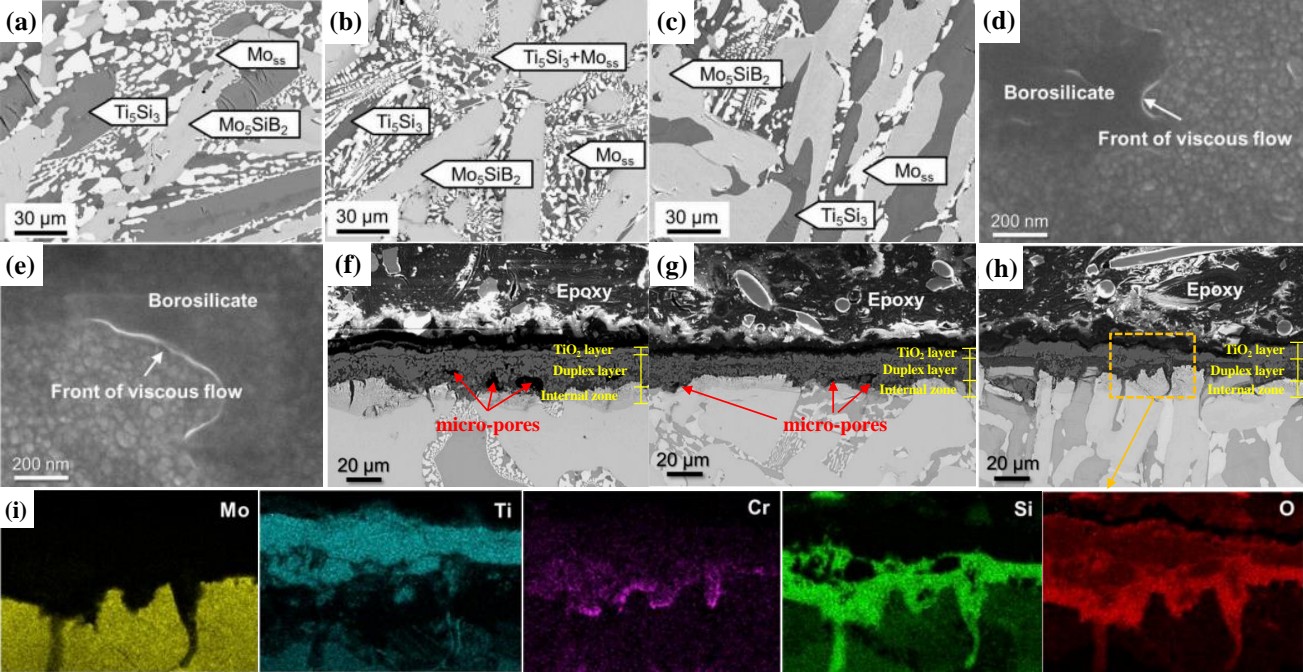

**Figure 18.** BSE images: (**a**) BASE, (**b**) AL and (**c**) CR; Surface SE images of different samples after oxidation 30 min at 700 °C: (**d**) AL and (**e**) CR; Cross-sectional BSE images of different samples after oxidation 24 h at 1100 °C: (**f**) BASE, (**g**) AL and (**h**) CR; (**i**) Elemental mappings of the micro-zone. Reproduced with permission [80]. Copyright 2020 Elsevier.

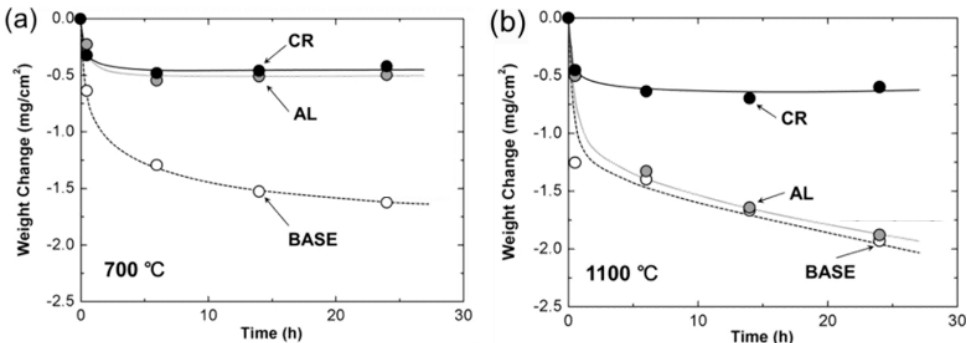

**Figure 19.** Weight change curves of different samples: (**a**) oxidized at 700 °C; (**b**) oxidized at 1100 °C. Reproduced with permission [80]. Copyright 2020 Elsevier.

### 3.3. Modification Mechanism of Metal Elements

According to the research on the microstructure and antioxidation performance of Mo-Si-B alloys containing metal elements, the modification mechanism of metal elements is obtained, as shown in Figure 20. The study reveals that the modification of the metal elements to the Mo-Si-B alloys is commonly achieved by changing the transient oxidation stage behavior of the alloy. As is seen from Figure 20b, the addition of Zr or Al cannot only refine the structure of the alloy, accelerate the formation of the borosilicate scale, but also play a "pinning" effect to inhibit the volatilization of molybdenum trioxide and internal diffusion of oxygen, thus improving the oxidation resistance of the alloy. Furthermore, the addition of an appropriate amount of Al can also promote the formation of dense $Al_2(MoO_4)_3$ or $Al_2O_3$ scale on the alloy surface, to passivate the alloy and prevent further oxidation. However, Zr or Al will compete with silicon and boron for oxidation, thus delaying the appearance of protective $B_2O_3$-$SiO_2$ film. Moreover, the oxidation of Zr or Al can form the large volume of oxide particles and distribute in the borosilicate scale. These particles will undergo phase transition at high temperature environments, leading to volume expansion or stress release. As a result, the oxide layer cracked and peeled off, and the oxidation resistance of the alloy deteriorated. Therefore, the addition of Zr or Al will show different modification effects on the antioxidation performance of the Mo-Si-B alloys, which may be related to preparation process, microstructure or B/Si ratio. Adding proper amount of Cr can form continuous protective $Cr_2(MoO_4)_3$ or $Cr_2O_3$ scale on the outer surface of $B_2O_3$-$SiO_2$ film, which can act as a diffusion barrier of oxygen, thus significantly enhancing the oxidative resistance of the alloy. The addition of a small amount of Fe can enhance the fluidity of borosilicate scale, and the $Fe_2(MoO_4)_3$ particles generated by oxidation fill in the gaps of the borosilicate scale and act as blocking phases. On the one hand, they prevent the internal diffusion of oxygen, on the other hand, they limit the volatilization of $MoO_3$, which have a positive effect on improving the antioxidation properties of the alloy. It can be observed from Figure 20c that the addition of Ti leads to the formation of thick and porous rutile-borosilicate duplex film on the surface of the alloy. In addition, the pores in the scale provide channels for the volatilization of molybdenum trioxide and inward spread of $O_2$, thus reducing the oxidative resistance of the alloy. The addition of Nb or W can destabilize the $Mo_3Si$ structure, thus improving mechanical properties such as high temperature compressive strength and fracture toughness of the alloy. However, Nb oxidizes to form coarse and porous $Nb_2O_5$ particles, which attach to the borosilicate scale and provide a fast route for the inward diffusion of $O_2$, resulting in the deterioration of the oxidation behavior of alloy. Moreover, the $WO_3$ particles formed by W oxidation are also embedded in the borosilicate scale, which destroys the integrity of the borosilicate scale and adversely affects the antioxidation of the alloy. However, as the oxidation temperature rises, the $WO_3$ will gradually volatilize, so W will not seriously damage the oxidation resistance of the alloy. This also reveals that the pretreatment of the alloy at higher temperatures can effectively improve its service life.

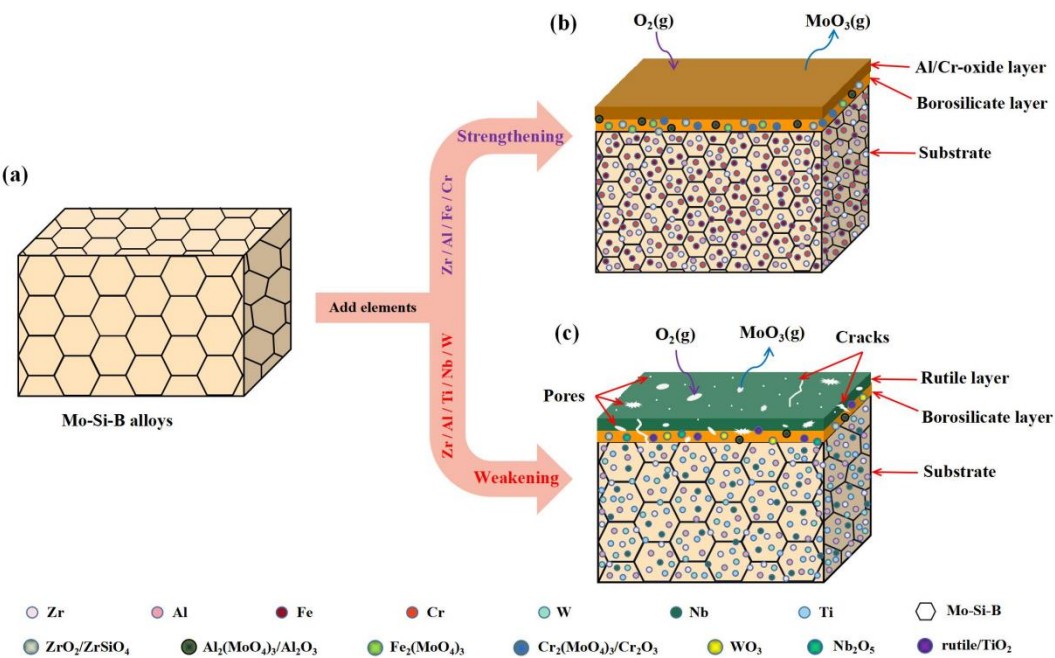

**Figure 20.** The schematic diagram of modification mechanism of metal elements. (**a**) Mo-Si-B alloys before oxidation (**b**) Strengthening (**c**) Weakening.

## 4. Conclusions

This work systematically studies the microstructure and oxidation behavior of Mo-Si-B ternary alloys. It is found that the antioxidation properties of Mo-Si-B ternary alloys are related to the processing method, grain siz,e and the content of silicon and boron elements. In the low silicon Mo-Si-B alloys, due to adding silicon and boron, the alloys form protective $B_2O_3$-$SiO_2$ films on the surface during oxidation process, so they have good oxidation resistance. Moreover, refining the microstructure of the alloy and moderately increasing the content of boron in the alloy can further enhance the antioxidation capacity of the low silicon Mo-Si-B alloys. By contrast, high silicon Mo-Si-B alloys are more prone to form continuous borosilicate films on the surface because of higher silicon content. Therefore, they exhibit better oxidation resistance than low silicon Mo-Si-B alloys.

Furthermore, the influences of metallic elements on the oxidation behavior of Mo-Si-B ternary alloys were also studied. It is shown that adding metallic elements can refine the grains and improve the microstructure of the Mo-Si-B alloys. Among them, adding elements such as Cr and Fe can significantly improve the antioxidation of Mo-Si-B alloys. In contrast, adding elements such as Ti, W and Nb will reduce the oxidation resistance of Mo-Si-B alloys. The modification effect of multiple metallic elements is usually more significant than single-metal modifications. It is worth noting that if the composition, preparation process or microstructure of the alloy are different, they may exhibit different oxidation behavior even if the same amount of Zr or Al elements is added. To further enhance the antioxidant effect of the alloy, the concentration of metal elements (such as Cr, Al) can be increased appropriately or the alloy can be pretreated at a higher temperature.

**Author Contributions:** The manuscript was written through the contributions of all authors. Y.Z.: conceptualization, investigation and supervision. Y.Z. and L.Y.: writing—original draft and image processing. L.Y., K.C. and F.S.: validation, resources, investigation, writing—review & editing. J.W., T.F. and X.Z.: visualization, writing—review & editing. All authors have read and agreed to the published version of the manuscript.

**Funding:** This work was supported by the Anhui Province Science Foundation for Excellent Young Scholars (2108085Y19) and National Natural Science Foundation of China (No. 51604049).

**Institutional Review Board Statement:** Not applicable.

**Informed Consent Statement:** Not applicable.

**Data Availability Statement:** Not applicable.

**Acknowledgments:** This work was supported by the Anhui Province Science Foundation for Excellent Young Scholars (2108085Y19) and National Natural Science Foundation of China (No. 51604049).

**Conflicts of Interest:** The authors declare no competing financial interest. The authors declare no conflict of interest.

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
