# Peer review of "Microstructure and Oxidation Behavior of Metal-Modified Mo-Si-B Alloys: A Review"

_coatings, doi:10.3390/coatings11101256_

Round 1
Reviewer 1 Report
The paper is an interesting dissertation on Oxidation behaviour of a Metal-Modified Mo- 2 Si-B Alloys taking into account the presence of phases
Just a few corrections must be done as suggested:
Row 32
industrial application. for example, ….
industrial application. For example, ….
Row 69
the silicon content is low, the researches of Mo-Si-B alloys are mostly concentrated
the silicon content is low, the researche of Mo-Si-B alloys is mostly concentrated
Row 88
After 100 h of oxidation, a dense and thick oxide scale formed on surface of the
After 100 h of oxidation, a dense and thick oxide scale formed on the surface of the
Row 101
annealing and laserl remelting,
annealing and laser remelting,
Row 123
enhance the antioxidation function of 123 the alloy.
enhance the antioxidant function of 123 the alloy.
Row 163
Figs. 5 (a-d) depicte the
Figs. 5 (a-d) depict the
Row 170
stable boroilicate scale,
stable borosilicate scale,
Row 305
alloy when the temperature below 1200 ℃. Especially, the oxidation speed of
alloy when the temperature is below 1200 ℃. Especially, the oxidation speed of
305 Mo-9Si-8B-1Zr (at.%) alloy was nearly three orders of magnitude slower than Mo-9Si-8B 306 sample, which revealed that the addition of Zr was instrumental in improving antioxida- 307 tion ability of Mo-9Si-8B composite, as depicted in Table 1. However, this beneficial effect 308 of Zr gradually disappeared when the temperature exceeded 1200 ℃. Fig. 10 (a) dis- 309 played the variation curve of the unit area mass with time of Mo-9Si-8B-(1Zr) samples 310 during oxidation at 1300 ℃. It could be observed that the oxidation trend of Row 316
of ZrO2 in the SiO2 scale occured, which
of ZrO2 in the SiO2 scale occurred, which
Row 404
rougher and existed a large number of holes, which maight be related to thermal chock
rougher and existed a large number of holes, which might be related to thermal shock
Row 422
the presence of Mo3Si can neither enhances the
the presence of Mo3Si can neither enhance the
Row 423
resistance nor improves the fracture toughness.
resistance nor improve the fracture toughness.
Row 548
continuous protection films generated on the surface continuous protection films are generated on the surface
Row 571
the addition of appropriate amount of Al
the addition of an appropriate amount of Al can also promote the formation of dense 571
Reviewer 2 Report
This review briefly summarizes the strategies adopted to improve the antioxidation performance of Mo-Si-B alloys. It can be accepted after revision. Please find the suggestions below.
- In row 97 and 98 on Page 3, how long does it take to achieve the steady-state oxidation for Mo-14Si-28B? It is better to compare the oxidation behavior between Mo-14Si-28B and Mo-12.5Si-25B?
- How to define the low silicon content or high silicon content of Mo-Si-B alloys?
- In Fig.4, the difference of oxidation rate between Mo-12Si-17B and Mo-12Si-8.5B seems negligible. Is there an optimum amount of B to achieve the best performance? Or the more B, the better the oxidation resistance?
- For high silicon content Mo-Si-B alloys, does the B content affect the formation continuous silica film?
- In section, what is the intrinsical difference between these two methods to prepare Mo-Si-B alloys? It is just because of the initial density?
- Could you please briefly introduce the advantages of the addition of W, Al and Ti? I can’t understand the addition of these harmful elements without other merits.
- For the beneficial element Cr, how to control its amount to achieve the desired performance?
Round 2
Reviewer 2 Report
The author has modified this manuscript according to the comments. I suggest it be accepted.